# Effects of Temperature on the Survival and Larval Development of *Deiratonotus Japonicus* (Brachyura, Camptandriidae) as a Biological Indicator

**Il-Kweun Oh *** and **Seung-Woo Lee**

Graduate School of Environmental Engineering, University of Kitakyushu, 1-1 Hibikino, Wakamatsu, Kitakyushu, Fukuoka 808-0135, Japan; leesw@kitakyu-u.ac.jp
* Correspondence: kweuni51@gmail.com; Tel.: +81-93-695-3293

**Abstract:** *Deiratonotus japonicus* (*D. japonicus*) inhabits isolated locations and upstream brackish waters from Kanagawa Prefecture to Okinawa Prefecture in Japan. This species faces the threat of extinction because of changing habitat conditions. Our previous studies have shown that its complete larval development from hatching to metamorphosis consists of five zoeal stages and one megalopal stage. In this study, the effect of temperature on the survival and growth of larval development in *D. japonicus* under controlled laboratory conditions of 13, 18, 23, 24, 25, and 26 °C was investigated by rearing larvae (30 PSU; 12:12 h light/dark cycle; fed a diet of *Brachionus plicatilis rotundiformis* and *Artemia* sp. nauplii). The survival rates and developmental periods were measured for each larval stage. The highest survival rates were obtained at 18–24 °C. Metamorphosis to megalopa occurred at 23–25 °C. There were rapid and synchronous developments at 25–26 °C but delayed and extended developments at 13 °C. The molting period decreased with increasing temperature. With decreasing temperature, the beginning of the development and duration of molting was prolonged. In addition, there were very low survival rates at 13 °C and 26 °C in all zoeal stages. Our results indicate that the early larval stages of *D. japonicus* are well adapted to 18–24 °C, the range observed in the estuarine marine environment of the Kita River during the breeding season. Optimum larval survival and growth were obtained at 23 °C. Temperature significantly affected the survival rate, developmental period, and molting of the larvae. The relationship between the cumulative periods of development from hatching through individual larval stages (*y*) and temperatures (*T*) was described as a power function ($y = a \times T^b$).

**Keywords:** *Deiratonotus japonicus*; Camptandriidae; larval stages; temperature; survival rate

## 1. Introduction

*Deiratonotus japonicus* (Decapoda: Camptandriidae), which was reported by Sakai for the first time in 1934 [1], is a rare crab species classified as near threatened in the Red Data Book List by the Ministry of Environment in Japan [2]. It is usually found in isolated locations and upstream brackish waters from Kanagawa Prefecture to Okinawa Prefecture in western Japan [1,3–6]. Previous studies have been limited to descriptions of the early zoeal stages [3], megalopal stages, habitat characteristics of *D. japonicus* [7], and genetic characteristics by Kawane et al. [8,9].

This species is used as a biological indicator of habitat quality in aquatic environments and is an important component of estuarine ecology. Despite its ecological importance, this crab species is presently under the threat of extinction because of the changing environmental conditions caused by pollution, habitat destruction, and natural disasters. Several factors control the distribution and diversity of this species, including changes in abiotic factors (temperature, salinity, substrate type,

and tidal currents) and biotic factors (food and food availability) associated with their habitat [10]. Moreover, in crustaceans with complex life cycles, their recruitment success, larval–juvenile survival, and growth may be affected by the variation in environmental conditions during embryonic and larval stages of development [11–13]; Gimenez and Torres, 2002).

Temperature is one of the most important environmental conditions for larval survival and crustacean development [14–19]. However, no studies have been reported on the effects of temperature on larval survival, development, and recruitment processes of *D. japonicus*. In this study, we investigated the optimum temperature condition for *D. japonicus* larvae by evaluating their survival rate and developmental period when maintained under different temperatures. The effects of temperature on the growth of early larval stages were investigated. However, our study not only provides information on optimal temperature conditions for these early larval stages but also presents new information for the conservation of *D. japonicus* in the Kita River.

## 2. Materials and Methods

The distribution of *D. japonicus* is limited to the upstream brackish areas (2.8–6.8 km) of the Kita River, Nobeoka, Japan (32°35′26″ N, 131°42′50″ E, Figure 1). The habitat of this species is sand and cobble sediment in water depths from 0 to 1.5 m. The annual water temperature ranges 10.5–24.3 °C and the tidal range is within 1.5 m. The mean salinity level ranges from 5 to 30 PSU. Reproduction in *D. japonicus* mostly occurs from late spring through late fall, with the maximum number of ovigerous females occurring in the summer. Egg color changes from bright orange to dark brown as the embryos mature.

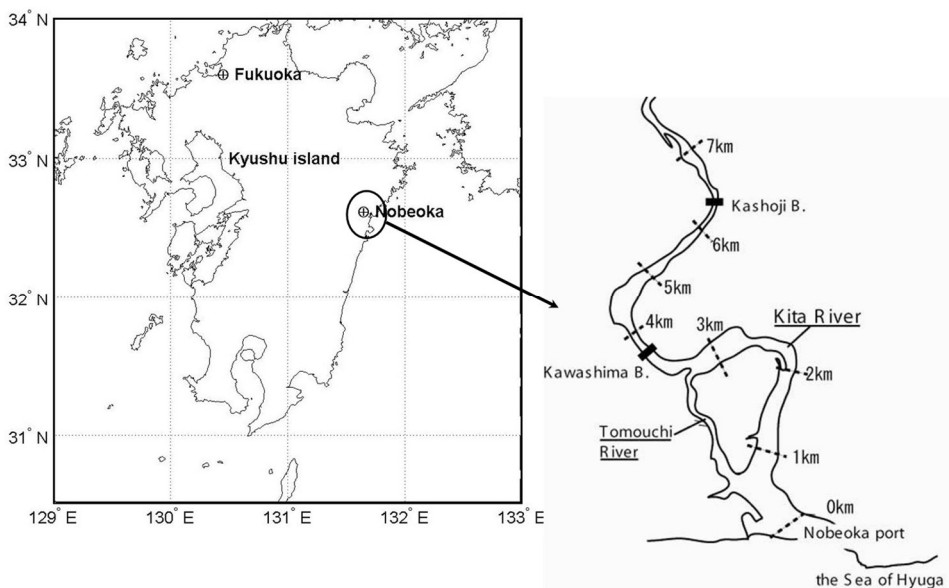

**Figure 1.** River estuary and sampling stations of *D. japonicus*.

Ovigerous *D. japonicus* females (i.e., with developing eggs) were collected from the Kita River by a quadrat trap with a net (0.5 × 0.5 × 0.3 m) 5.75 km upstream. The ovigerous females were transported in aerated 3-L containers to the laboratory. Individual ovigerous females were maintained from egg laying to hatching of the first zoeal stage under controlled conditions of temperature (20 °C), salinity (4 PSU), and photoperiod (12:12 h light/dark cycle) in 2-L water tanks. The presence of newly hatched larvae in the rearing tanks was checked at 12 h intervals.

Larvae usually hatched at night. Immediately after hatching, actively swimming larvae were collected with wide-bore pipettes after being induced to concentrate in one corner of the rearing tank where there was a shaft of light. They were individually transferred to a rearing tank filled with 2-L of filtered seawater (membrane filter of 0.45 μm, 30 PSU) at 20 °C. The larvae were acclimated to additional

constant temperatures of 13, 18, 23, 24, 25, and 26 °C at a rate of 2–3 °C/h by transferring rearing tanks suspended in trays between water baths set at the appropriate temperature. The larvae were fed daily with *Brachionus plicatilis rotundiformis* at a rate of 50 indiv/mL after hatching to the fifth zoeal stage. *Artemia* sp. nauplii were added at an increasing density of approximately 2–5 indiv/mL from the second zoea up to the fifth zoeal stage. When the larvae metamorphosed to megalopa, the density of *Artemia* sp. nauplii was increased to 10 indiv/mL. In each temperature test, two experimental replicates of the first zoeal stage, 100 larvae per replicate, were reared. Slight aeration provided oxygen saturation and sufficient turbulence to prevent the settling of food organisms. Larvae and prey were transferred to newly prepared rearing tanks with new filtered seawater (using the same membrane filter size of 0.45 μm, 30 PSU) using a wide-bore pipette each morning. After transfer, dihydrostreptomycine sulfate (Wako, Osaka, Japan) was added to the rearing water at 50 mg/L to prevent from bacteria attaching to the larvae [20]. The larvae were observed daily. During transfer, the survival and larval stages were recorded. Different zoeal stages were easily distinguished by their appendages and zoea and megalopa were reared in separate tanks to avoid cannibalism. Daily analyses continued until all larvae had either metamorphosed to the megalopal stage or died. Larvae were considered dead when no visible movement of internal or external appendages could be detected under a dissecting microscope. Larval stages of dead larvae were also recorded.

Our statistical analyses followed those reported by Sokal and Rohlf [21]. A one-way ANOVA followed by comparisons between pairs of means was used to compare survival and periods of development for each larval stage [22]. To determine whether temperature significantly influenced the survival rate, stage-specific and cumulative duration of development were compared with temperature treatments using a one-way ANOVA. Tukey's range test or Duncan's new multiple range test was used to compare the differences identified by the ANOVA. Percentage data were normalized by using an arcsine transformation for statistical analysis. Differences between treatments were considered significant at $\alpha = 0.05$ for all statistical tests. The duration of individual stages, as well as cumulative periods of development, were described as non-linear power functions of temperature [16,23,24]. All statistical analyses were conducted using the XLSTAT statistical package Ver. 10.04 (Addinsoft, NY, USA).

## 3. Results

### 3.1. Survival Rates

The survival rates of each larval stage reared under different temperature conditions are presented in Table 1. Changes in the survival rates by larval stage with time after hatching are shown in Figures 2 and 3. The first zoeal stage reared at 13 °C did not reach the fourth zoeal stage, even though survival rates to the first zoeal stage were between 87.0% and 93.5% at 18-24 °C. This indicates that temperature becomes a significant parameter for the longer survival of zoeal. All larvae died 32-38 days after hatching. Of those larvae reared at 23 °C, 4.5% survived to megalopa. Larvae completed development to megalopa at 23, 24, and 25 °C with survival rates of 0.5%-4.5%. However, individuals that reached the megalopal stage under any of the temperature conditions were not able to molt successfully to the first crab stage.

**Table 1.** Mean survival rates (% of survivors for a given stage) of *D. japonicus* to reach each larval stage reared under different temperature conditions.

| Temp | Zoea I | | Zoea II | | | | Zoea III | | | | Zoea IV | | | | Zoea V | | | |
|---|---|---|---|---|---|---|---|---|---|---|---|---|---|---|---|---|---|---|
| °C | % | ±SD | % | ±SD | Cum %[2] | ±SD | % | ±SD | Cum %[2] | ±SD | % | ±SD | Cum %[2] | ±SD | % | ±SD | Cum %[2] | ±SD |
| 13 | 51.2[b] | 3.0 | 57.8 | 2.1 | 29.5[b] | 0.7 | | | | | | | | | | | | |
| 18 | 89.4[a] | 3.8 | 79.2 | 14.1 | 71.0[a] | 15.6 | 72.5 | 7.6 | 50.9[ab] | 5.9 | 67.6 | 4.7 | 34.5[a] | 6.4 | | | | |
| 23 | 93.5[a] | 2.1 | 77.1 | 3.3 | 72.0[a] | 1.4 | 83.4 | 0.4 | 60.0[a] | 1.4 | 76.6 | 5.2 | 46.0[a] | 4.2 | 2.1 | nd[3] | 1.0[a] | nd[3] |
| 24 | 87.0[a] | 14.1 | 77.8 | 18.2 | 69.0[a] | 26.9 | 72.6 | 17.8 | 52.5[ab] | 31.8 | 73.7 | 13.6 | 36.5[a] | 16.3 | 11.3 | 4.7 | 4.5[a] | 3.5 |
| 25 | 72.2[ab] | 15.8 | 62.0 | 14.1 | 43.7[ab] | 0.5 | 43.6 | 3.8 | 19.0[bc] | 1.4 | 33.4 | 23.5 | 6.5[b] | 5.0 | 16.7 | nd[3] | 0.5[a] | nd[3] |
| 26 | 50.0[b] | 14.1 | 59.1 | 4.9 | 29.2[b] | 5.9 | 50.0 | 14.1 | 14.2[c] | 1.2 | 50.9 | 34.2 | 7.0[b] | 4.2 | | | | |

[1] Mean ± SD of the mean survival rate (%) to reach the larval stage by each temperature; [2] Cumulative survival from hatching through a given stage (% of initial number at hatching); [3] No SD (*n* = 1); [a, b, c] Significant differences were found between groups with different superscripted letters in the same column (*p* < 0.05).

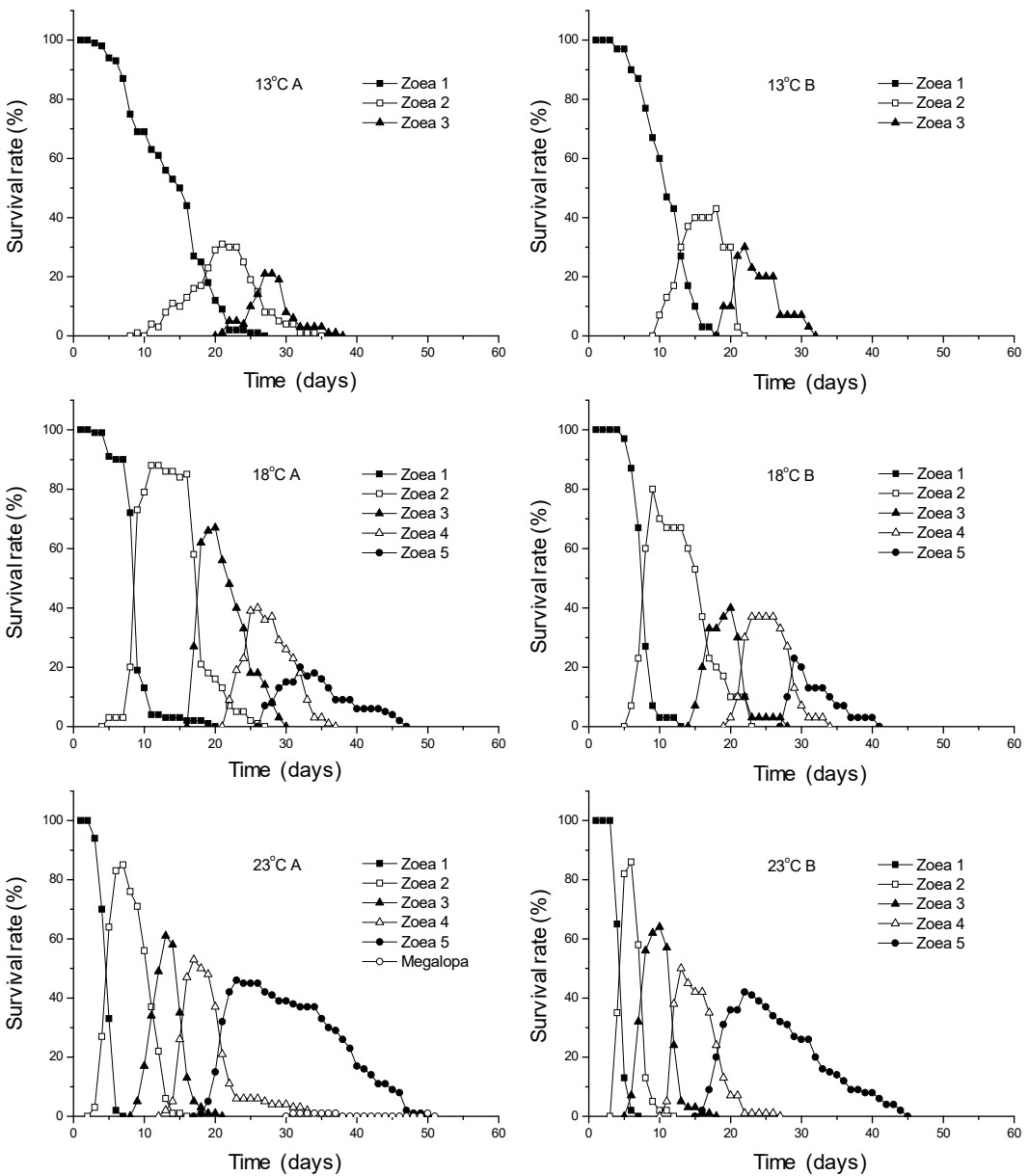

**Figure 2.** Survival rate (%) for each developmental stage from hatching of *D. japonicus* reared at 13, 18, and 23 °C.

Mass mortality occurred during the metamorphosis to the zoeal stage at 13, 25, and 26 °C. Mortality of the larval stages was strongly associated with molting (Table 1, Figure 3). There was rapid and synchronous development at 25 °C and 26 °C that were delayed and extended at 13 °C. With increasing temperature, mortality and molting occurred continuously. The results show that temperature significantly affected the survival of the early larvae of *D. japonicus*. The survival rates were significantly different ($p < 0.05$) between 23 °C and 13 °C and 26 °C, whereas there were no significant differences ($p > 0.05$) among survival rates at 18–24 °C, showing relatively high survival rates. The best survival rate was observed at 23 °C (Figures 2 and 3).

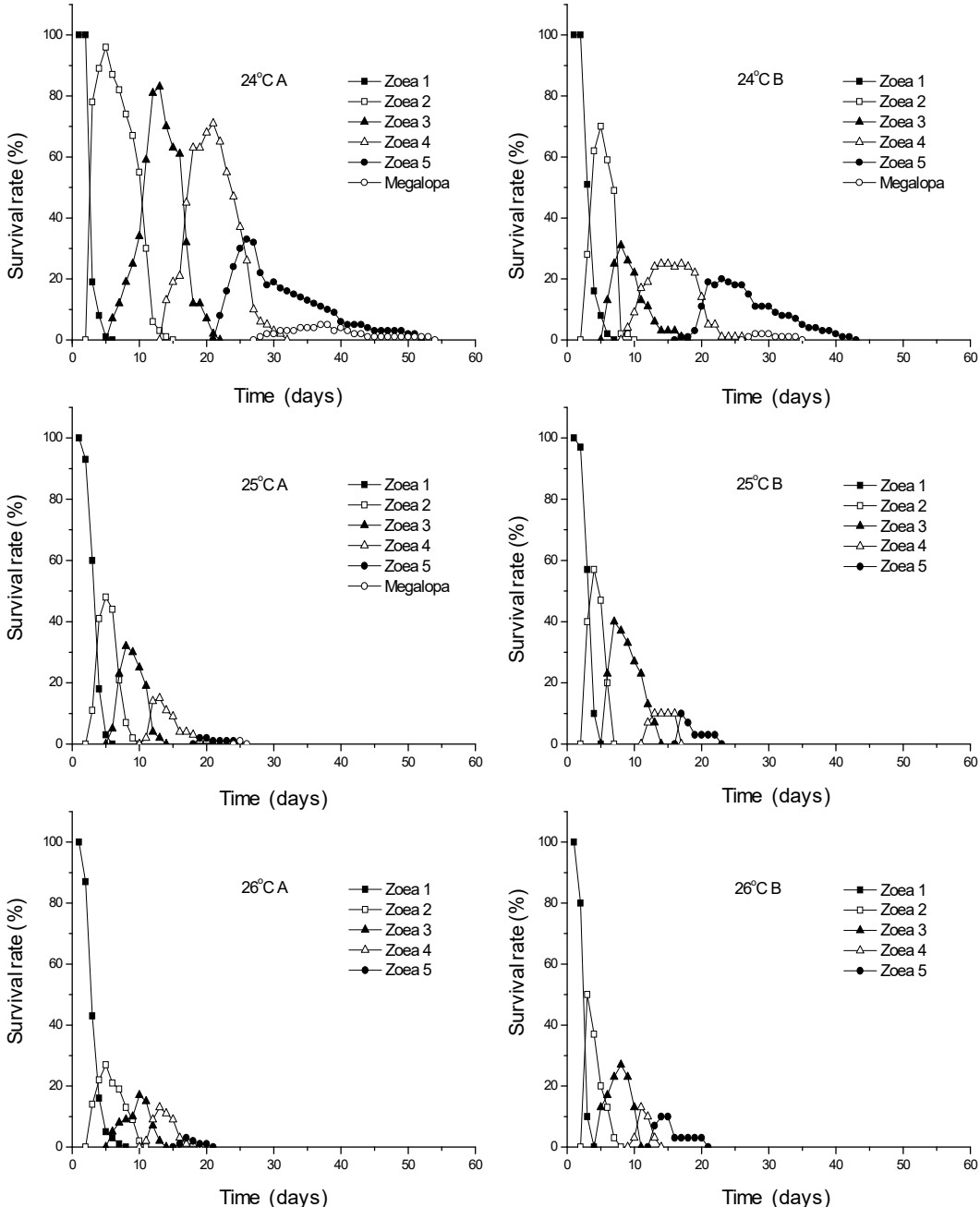

**Figure 3.** Survival rate (%) for each developmental stage from hatching of *D. japonicus* reared at 24, 25, and 26 °C.

## 3.2. Growth and Development

The development periods of individual larval stages are given in Table 2. Growth in each successive developmental period decreased with increasing temperature (Table 2, Figure 2). In addition, molting was delayed and extended at 13 °C. Developmental periods were significantly longer ($p < 0.05$) at 13 °C and 18 °C than 23 °C and 26 °C, and the third zoeal stage was delayed by 6–8 days. The duration of larval development from the first zoea to the fifth zoeal stage required a minimum of 14–19 days at 23–26 °C. There were no significant differences in average periods to the megalopal stage when reared at 23–25 °C. A shorter molting period was significantly associated with increasing temperature. Developmental periods taken to reach the megalopal stage, as well as individual zoeal stages, were inversely related to temperature (Table 2, Figure 4). The average duration until the megalopal stage ranged from 23 to

27 days at 23 to 25 °C. Larval molting also synchronized rapidly at 25–26 °C. The megalopa did not metamorphose to the first juvenile crab stage in all experiments.

**Table 2.** Mean developmental period (days) from hatching required to reach each larval stage of *D. japonicus* reared under different temperature conditions.

| Temp °C | Days for Each Developmental Stage [1] | | | | |
|---|---|---|---|---|---|
| | **Zoea II** | **Zoea III** | **Zoea IV** | **Zoea V** | **Megalopa** |
| 13 | 8.0 ± 0.0 [a] | 10.5 ± 2.1 [a] | | | |
| 18 | 4.0 ± 0.0 [b] | 10.5 ± 2.1 [a] | 5.0 ± 0.0 [a] | 6.5 ± 2.1 [ab] | |
| 23 | 2.5 ± 0.7 [c] | 4.0 ± 2.8 [b] | 4.5 ± 0.7 [a] | 5.0 ± 0.0 [ab] | 13.0 [2a] |
| 24 | 2.0 ± 0.0 [c] | 3.0 ± 0.0 [b] | 5.5 ± 3.5 [a] | 8.0 ± 0.0 [a] | 8.0 ± 2.8 [a] |
| 25 | 2.0 ± 0.0 [c] | 3.0 ± 0.0 [b] | 5.5 ± 0.7 [a] | 6.5 ± 2.1 [ab] | 6.0 [2a] |
| 26 | 2.5 ± 0.7 [c] | 2.5 ± 0.7 [b] | 5.0 ± 0.0 [a] | 4.0 ± 1.4 [ab] | |

[1] Mean ± SD shows the mean number of days taken to reach larval stage by each temperature; [2] No SD ($n = 1$); [a, b, c] Significant differences were found between groups with different superscripted letters in the same column ($p < 0.05$).

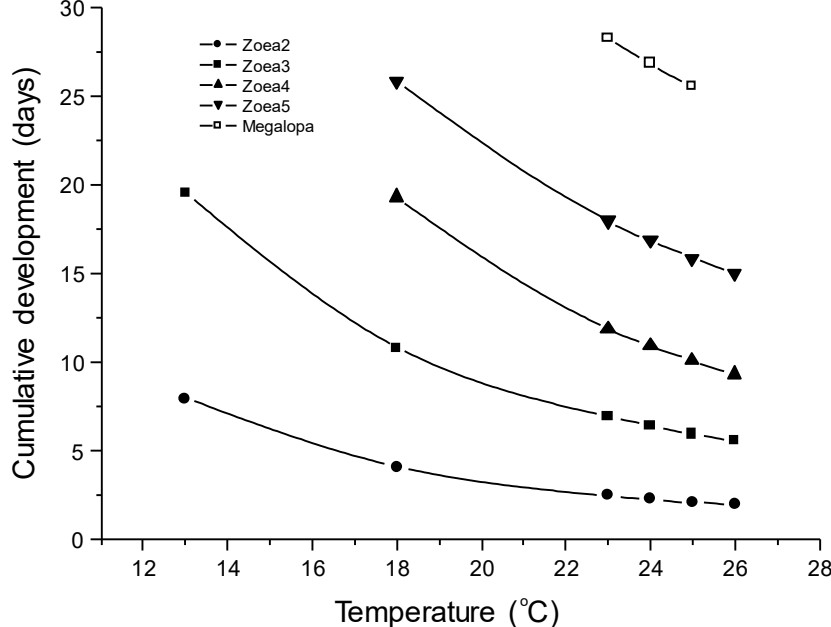

**Figure 4.** Relationship between temperature and developmental period from hatching to reach each larval stage of *D. japonicus*. Curves show theoretically calculated values using non-linear regression equations (power functions, $y = a \times T^b$).

The models of the developmental period (D, in days) in relation to temperature (T, in °C) could be described as a power function, $D = a \times T^b$, where *a* and *b* are fitted parameters. All curves of non-linear regressions are illustrated in Figure 4. Non-linear regressions of cumulative development times in relation to temperature are shown in Table 3 by individual developmental stage with complete successive larval stages over six temperature changes. The cumulative development times taken to reach megalopa as well as individual stage developmental periods were inversely related to temperatures (Table 1, Figure 4). Larval developmental periods for each stage showed similar trends over all temperatures. All determination coefficients ($r^2$) were ≥ 0.897. The best fit between the observed and predicted data ($r^2 > 0.995$) was found in the cumulative time of development from hatching to megalopa (Table 3). The curves of these regressions were not significantly different between each zoeal and megalopal stage (ANOVA, $p > 0.05$) indicating generally similar developmental stages.

**Table 3.** Fitted parameters (*a, b*) and determination coefficients ($r^2$) of non-linear regression equations (power functions, $y = a \times T^b$) describing the development time (*y*, day) as a function of temperature (T, °C). Cum. dev: cumulative time of development from hatching to later stages.

| Parameter | Zoea II | Zoea III | Zoea IV | Zoea V | Megalopa |
|---|---|---|---|---|---|
| Cum. dev | | | | | |
| a | 1436.9 | 2044.6 | 5938.7 | 1903.4 | 1198.5 |
| b | −2.026 | −1.814 | −1.982 | −1.487 | −1.195 |
| $r^2$ | 0.985 | 0.898 | 0.975 | 0.897 | 0.995 |

## 4. Discussion

Recruitment is an important process in crustaceans with complex life cycles and is determined by the physical and biological processes that occur during the pelagic larval stage, settlement, and in juvenile stages [25]. The pelagic larvae of many estuarine species, such as brachyuran crabs, are transported away from adult habitats by selective stream tidal transport. Brachyuran larvae develop for approximately 1–2 months in the lower regions of bays, estuaries, and the continental shelf and must be transported back to the adult habitats [26–29]. Furthermore, these unique characteristics may result in differences in their distribution and life cycle strategies.

Reproductive larval release among decapod crustaceans is often synchronized with natural cycles such as the lunar phase, tide, and time of day [30–33]. Most larval releases occur during the night and on a flood tide to reduce their vulnerability to predators and low salinities and to facilitate their transport to their appropriate nursery habitats [31–33]. In our experiments, *D. japonicus* larval releases were mostly observed to occur at night.

In the Kita River, reproduction in *D. japonicus* mostly occurs from late spring (April) until late fall (November) with recruitment peaks of ovigerous females in the summer. The geographic distribution of marine organisms is chiefly determined by temperature and salinity. Usually, larvae tolerate narrower ranges than adults of the same species do. Commonly, the tolerance of larval stages to temperature is narrower than those of adults. Adult populations of *D. japonicus*, including ovigerous females, were found in upstream brackish waters of the Kita River from 2.8 to 6.8 km, where the annual temperature ranges from 10.5 to 24.3 °C. Areas from 2.8 to 5.75 km in the Kita River are characterized, at least seasonally, by a higher relative abundance of early juvenile *D. japonicus*. This inferred that the newly hatched larvae of *D. japonicus* aggregated in surface waters during a flood tide at night. After a three- to four-week period of growth and development in the open sea and estuaries, the megalopa molt and they settle out as juvenile crabs in upper intertidal zones. Figure 5 shows the geographical temperature distribution in the Kita River.

The survival and longevity of marine invertebrate larvae are influenced by abiotic factors such as temperature and salinity, and by biotic factors such as food availability, food, and predation [34]. Temperature is known to affect the survival and development of larvae and breeding activity in many decapod crustaceans [14–17,24,35–37]. In many brachyuran species, temperature affects larval survival and the duration of development, and a decrease in larval duration with increasing temperature has been widely reported for these species [37–43]. Furthermore, the risk of predation and physical stress in pelagic environments is reduced when larval development is abbreviated. Shirley et al. showed that the growth, morphology, and survival of crustacean larvae were influenced by temperature [38]. Brown et al. investigated temperature-salinity effects on the survival and development of the early larval stages of *Menipp mercenaria* [14]. Temperature affects the rate of embryonic and larval development from zoeal stages through to the early crab stages of the blue swimming crab, *Portunus pelagicus* [39]. Dawirs [40] and Newman et al. [16] used developmental duration and water temperature regression models to predict the larval period of decapods in the natural environment. Additional studies of larval behavior indicate that zoeal stages are extremely sensitive to changes in temperature, salinity, and pressure and respond by moving vertically up the column. These upward movements of brachyuran

larvae are associated with a decrease in temperature and increase in salinity [41–43]. This suggests that tidal vertical migration is worth investigating in the pelagic larvae of coastal invertebrates as a possible mechanism for assisting dispersal or recruitment.

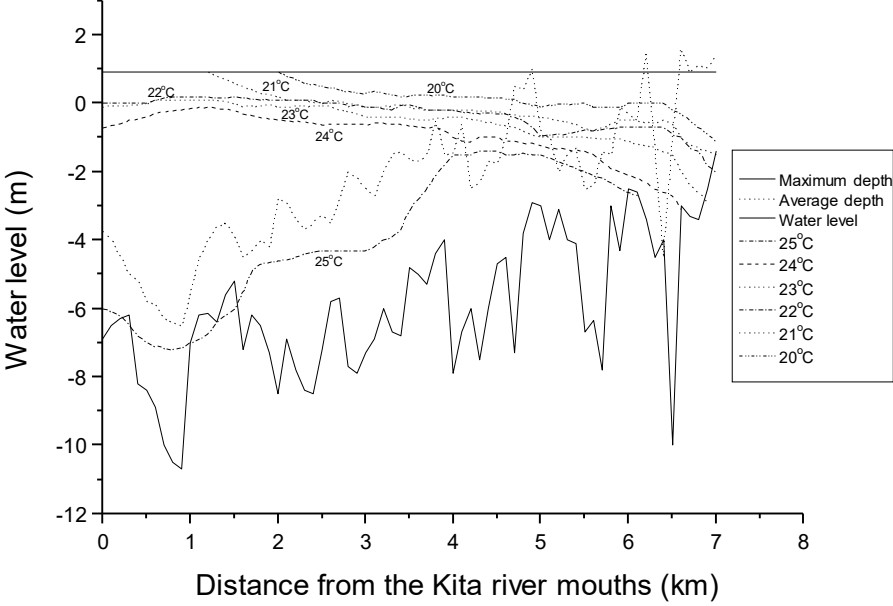

**Figure 5.** Water temperature distribution on a spring tide in the Kita River.

Most Camptandriidae crab species, such as *D. japonicus*, are classified as having a pelagic marine mode of development. They produce many small eggs and the females release pelagic larvae in coastal marine waters. Their development is usually comprised of four to five zoeal stages and a megalopal stage. Larval development of *D. japonicus* in the laboratory was described by Terada [3] and Hiu et al. [44] and consists of the five zoeal stages and a megalopal stage. Our results agreed with those of Hiu et al. and we can compare our findings with previous accounts of *D. japonicus* larval durations. Terada recorded an average time to the second zoeal stage of 7–8 days at 18.8–22 °C, compared with 2.5–4.0 days in the present study [3]. Hiu et al. recorded an average time to megalopa of 38 days at 21.0–22.8 °C [44], compared with 29 days at 23.0 °C in the present study. Thus, despite considerable variation in experimental procedures, developmental periods are very comparable. Our results showed that *D. japonicus* zoeal developmental periods were significantly longer at 13 °C and significantly shorter at 25–26 °C. Generally, an increasing temperature had a significant accelerating effect on development, especially in the high-temperature range. The effect of temperature on zoeal developmental periods can impact recruitment success and these developmental periods can affect dispersal, survival, and growth based upon predation, food availability, and bioenergetics. Moreover, estimates of larval dispersion ranges and mortality due to predation will also be influenced by the pattern of temperatures to which larvae are exposed throughout their developmental process.

Those in the first zoeal stage did not reach the fourth zoeal stage at 13 °C, while larvae reared at temperatures of 23–25 °C completed their development to megalopa [45]. The survival rates ranged from 0.5% to 4.5% (Table 1) and no megalopa was able to molt successfully to become a first juvenile crab. Zoeal survival rates were reduced when reared at 13, 25, and 26 °C, and these temperatures were probably outside of the optimal range for larvae of *D. japonicus*. With increasing temperature, mortality and molting occurred synchronously. This was shown by a significantly low survival rate with increasing temperatures (Table 1 and Figures 2 and 3). It is also possible that the decline in feeding at low temperatures contributed to the decreased survival rate of the larval stages. The survival rates of *D. japonicus* were significantly higher at 18–24 °C than at 13, 25, and 26 °C, and the lowest survival rate was at 26 °C. The optimal survival rate to the megalopal stage occurred at 23 °C.

We could not complete the larval development of *D. japonicus* in the laboratory as no megalopa reached the first juvenile crab stage in our experiments. The reason for the 100% mortality of megalopa remains unsolved. Perhaps this stage might prefer more brackish water (salinity stress) or a particular substrate for molting. Feeding behavior and food are also different between the zoeal and megalopal stages. Unsuitable food items may decrease the survival rate or increase the duration of the megalopal stage. In the future, these hypotheses should be applied to studies of *D. japonicus* megalopal stage, which should be more focused on the conditions of temperature, salinity, substrate, and food to limit mortality effects and the duration of development. These experimental results will further clarify the distribution pattern and complete life cycle of *D. japonicus* that present high ecological relevancy in the Kita River. Furthermore, these findings are important for defining management strategies and the conservation of *D. japonicus*.

## 5. Conclusions

In this study, we showed that temperature significantly affected the survival rate, developmental periods, and molting of larval stages in *D. japonicus*. Survival rates of larvae in the zoeal stage were generally high at 23–24 °C. With increasing temperature, there were rapid and synchronous developments at 25–26 °C but delayed and extended development at 13 °C. Our results indicate that the early larval stages are well adapted to 18–24 °C. These results confirm that the optimal rearing temperature in terms of larval survival and growth is 23 °C. These temperature conditions are usually found in the estuarine marine environment of the Kita River during the breeding season. Furthermore, as with the estuarine brachyuran crab, these results support the assumption that newly hatched larvae of this species are mainly transported to the lower regions of bays, estuaries, or coastal marine waters for larval development and that reinvasion to adult habitats occurs by the megalopal stage.

**Author Contributions:** I.-K.O. planned the experiment and collected and analyzed data. I.-K.O. prepared the manuscript and S.-W.L. helped in drafting the manuscript and interpretation of the results. All authors contributed to revisions and completion of the manuscript. All authors have read and agreed to the published version of the manuscript.

**Funding:** This research received no external funding.

**Acknowledgments:** This project was supported by the River Ecology Group of Japan. We also thank the staff of the School of Ocean Fisheries and Sciences at Kyushu University of in providing *Brachionus plicatilis rotundiformis*.

**Conflicts of Interest:** The authors declare no conflicts of interest. We have already mentioned funding sources in acknowledgment section; no other relationships or activities that could appear to have influenced the submitted work.

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
