# Peer review of "Effects of Temperature on the Survival and Larval Development of Deiratonotus Japonicus (Brachyura, Camptandriidae) as a Biological Indicator"

_jmse, doi:10.3390/jmse8030213_

Round 1

Reviewer 1 Report

First off there some minor edits that need fixing.  Scientific names need to be in italics (lines 10, 29, 77, 78 and 80) and the font is different on lines 36, 37, 112 and 113. The general premise and results are that temperature effects development timing and those are consistent with the literature.  The survival rates to megalopa stage are low and it is disappointing that less is discussed about mortality rates and why there was no survival of megalopa stage individuals.  There is no citation for the information on timing of recruitment (lines 247 and 248), which should have a source for that information.  The biology described is interesting and logical for this species, but maybe using some more salinity variation might have helped since this is an estuarine species or at least that issue could be discussed relating to results.  The authors could also be a bit more specific about how their results would help in management of this endangered crab species.  The text is clearly written and is a positive contribution, but could use some more specific discussion as outlined.

Author Response

Please find the attached file including the response to Reviewer's comments.

Reviewer 2 Report

THis study is very serious, easy to read and well presented. Although I am not a native English speaker, I found no error (usually I find some when readers are not native).

-Very sound research explained with many adequate details (experimental rpocedure well described and well designed).

However the discussion is repetitious with results at several places.

I am not sure that it is clear in the results that megalopes died before molting. Are data clear about that ? I read the paper rapidly (I admit) but I had the feeling that maybe th experiment had not been pursued long enough for them to hatch in juvéniles. PLease check that it is clear that all larvae died when the experiment ended. If necessary, precise it more clearly (but maybe it is my fault).

L108: MAybe you meant "third zoeal stage" here, not "first" ?

L201: could you give references for this? Maybe you placed them elsewhere in the text but ... here it would be nice.

L242 and above (also below): in this part of the discussion, results are repeated which makes reading uneasy and lengthens the text. The discussion should be checked and repetitions of results should be avoided.

L243: this has been writtne at severalplaces. NO nead to repeat again and again.

Author Response

(The authors gave the same response as above.)
